# Omentin—General Overview of Its Role in Obesity, Metabolic Syndrome and Other Diseases; Problem of Current Research State

**DOI:** 10.3390/biomedicines13030632

**Published:** 2025-03-05

**Authors:** Hubert Mateusz Biegański, Krzysztof Maksymilian Dąbrowski, Anna Różańska-Walędziak

**Affiliations:** 1Medical Faculty, Collegium Medicum, Cardinal Stefan Wyszynski University in Warsaw, 01-938 Warsaw, Poland; ski.bieg.an@gmail.com (H.M.B.); krzmaksdab@gmail.com (K.M.D.); 2Departament of Human Physiology and Pathophysiology, Faculty of Medicine, Collegium Medicum, Cardinal Stefan Wyszynski University in Warsaw, 01-938 Warsaw, Poland

**Keywords:** omentin, intelectin-1, obesity, metabolic syndrome, diabetes mellitus, atherosclerosis, hypertension, inflammation

## Abstract

**Background:** Omentin (omentin-1, intelectin-1, ITLN-1) is an adipokine considered to be a novel substance. Many chronic, inflammatory, or civilization diseases are linked to obesity, in which omentin plays a significant role. **Methods**: MEDLINE and SCOPUS databases were searched using the keywords “omentin” or “intelectin-1”. Then the most recent articles providing new perspectives on the matter and the most important studies, which revealed crucial insight, were selected to summarize the current knowledge on the role of omentin in a literature review. **Results and Conclusions**: The valid role of this adipokine is evident in the course of metabolic syndrome. In most cases, elevated omentin expression is correlated with the better course of diseases, including: type 2 diabetes mellitus, polycystic ovary syndrome, rheumatoid arthritis, metabolic dysfunction-associated steatotic liver disease, Crohn’s disease, ulcerative colitis, atherosclerosis, or ischemic stroke, for some of which it can be a better marker than the currently used ones. However, results of omentin studies are not completely one-sided. It was proven to participate in the development of asthma and atopic dermatitis and to have different concentration dynamics in various types of tumors. All of omentin’s effects and properties make it an attractive subject of research, considering still unexplored inflammation mechanisms, in which it may play an important role. Omentin was proven to prevent osteoarthritis, hepatocirrhosis, and atherosclerosis in mouse models. All of the above places omentin among potential therapeutic products, and not only as a biomarker. However, the main problems with the omentin’s research state are the lack of standardization, which causes many contradictions and disagreements in this field.

## 1. Introduction to Omentin—History and Structure

Omentin-1, also known as intelectin-1 (ITLN-1), is one of the two proteins belonging to the human intelectins group. For the first time, interlectin-1 was described in oocytes and embryos of *Xenopus laevis* in 1986. It was then named XL35 and thought to play a role in morphogenesis [1,2]. In the year 1998, intelectin-1 was isolated from intestinal Paneth cells of a mouse and called intelectin, as for intestinal lectin. The researchers associated this protein’s function with immune response against intestinal microorganisms [3]. Finally, in 2001, intelectin was discovered in a human body and called human intelectin (hIntL) [4], nowadays known as omentin. Omentin-1 owes its final name to its identification from expressed sequence tags (ESTs) of a human omental fat cDNA library in which it is more prevalent than most other known adipokines [5].

Omentin is an adipokine, which is a peptide hormone whose expression takes place primarily in adipose tissues [5]. It consists of 313 amino acids and has an approximate molecular weight of 35 kDa. It comes in the homotrimeric form, and its structure highly resembles other mammals’ intelectin in up to 80%. Different species, however, may have other main sites of gene expression or its intensity [4,6]. Omentin is encoded on chromosome 1q21.3 [7,8,9]. Interestingly, adipocytes themselves are not involved in the process of omentin’s secretion. This role belongs to surrounding cells in human adipose tissue [5,10]. Synthesis of omentin is expressed much more in visceral adipose tissue (VAT) than in subcutaneous adipose tissue (SAT) [11]. It is also present in many other locations, such as endothelial cells, Paneth cells, goblet cells, in lungs or epicardium of the heart, and many others [12,13].

## 2. Results

### 2.1. Omentin—Mechanism of Its Action, Function, and Relationship with Various Conditions

Omentin has been gaining popularity as a research topic over the last 20 years. The interest in it has increased substantially since its connection with numerous diseases, especially those associated with the metabolic syndrome, was revealed. Metabolic syndrome is a characteristic comorbidity of at least 3 risk factors for cardiovascular diseases and diabetes mellitus, including: abdominal obesity, high plasma concentration of triacylglycerols (TAG), low plasma concentration of high-density lipoprotein (HDL), high blood pressure (BP), and hyperglycaemia on an empty stomach [14].

Various studies identified in vitro effects of omentin and observed differences in its levels depending on the patient’s condition. Research conducted on isolated adipocytes showed that omentin’s core function, prominent even at sub-physiological concentrations, is a temporary increase in cells’ insulin sensitivity and promotion of insulin-induced glucose uptake from blood. However, omentin does not increase basal cell glucose uptake, in contrast to many other adipokines. It should be noted that omentin has a greater influence on SAT, which constitutes approximately 80% of human adipose tissue, than on VAT [5]. Researchers’ attention has also been brought to the proximity of the gene encoding omentin to genes previously associated with the occurrence of type 2 diabetes mellitus (T2DM) [15,16]. Nevertheless, no significant correlation between these genes and T2DM has been demonstrated so far [17,18,19].

An important observation is the inverse correlation between T2DM and obesity and lower levels of omentin gene expression [14,20,21,22]. As a result, a number of studies connected the role of omentin with many disorders, such as: polycystic ovary syndrome (PCOS) [23], rheumatoid arthritis (RA) [24], metabolic dysfunction–associated steatotic liver disease (MASLD), previously known as non-alcoholic fatty liver disease (NAFLD) [25], atherosclerosis [10], and Crohn’s disease (CD) [7]. It has also been noted that omentin prevents excessive vasoconstriction in response to norepinephrine (NE) by inducing nitric oxide (NO) synthesis in epithelial cells [26]. Omentin prevents angiogenesis in vitro [27] and has an anti-inflammatory effect due to reduced c-reactive protein’s (CRP) and NF-κB expression induced by omentin [28]. All this data indicate a significant role of omentin in regulating systemic inflammation and not only the glucose metabolism.

### 2.2. Omentin in Obesity, Metabolic Sydrome and Comorbidites

#### 2.2.1. Obesity, Overweight, and Plasma Omentin Correlations

Overweight and obesity, commonly understood as excessive accumulation of adipose tissue in the human body, are diagnosed in patients whose body mass index (BMI) exceeds 25 and 30 kg/m^2^, respectively [29]. Excessive body mass is a problem that has been building up since the 1960s and 1970s to the point where nowadays it is referred to as a global epidemic [30]. In 2022 approximately 43% of all adults worldwide were struggling with overweight and 26% with obesity. These statistics are even less favorable in developed countries; for instance, in the United States, as much as 67% of its population is overweight [29]. This condition is associated with numerous health risks: increased risk of cardiovascular diseases, diabetes mellitus, cancer, and chronic lung diseases [29,31]. For that reason, determining the function of omentin became an attractive research path for uncovering potential mechanisms behind obesity and diabetes mellitus and moving closer to developing treatment methods for them.

A negative correlation between the plasma omentin concentration and increased body mass was reported in numerous research papers that took into consideration BMI, plasma leptin concentration, and waist circumference [14,32,33]. In overweight and obese subjects, the value of these parameters was in an inverse proportion to the plasma omentin concentration. This data dependency was observed both in adults [14,32] and in children [33]. A decrease in the omentin-encoding gene expression was also observed in visceral adipose tissue of overweight individuals. The reason behind these correlations has not yet been established. It is suspected that this is related to significantly increased pro-inflammatory factors like tumor necrosis factor-α (TNF-α) and interleukin-1 (IL-1) in obesity [34].

#### 2.2.2. Type 1 and 2 Diabetes Mellitus, Insulin Insensitivity, and Microcirculation Damage

Furthermore, in patients with T2DM, the development of which is due, among other things, to excessive accumulation of adipose tissue [35], the plasma omentin concentration is also reduced. This was confirmed in individual studies taking into account plasma concentration of glucose, insulin resistance index HOMA-IR [32], as well as in cross-sectional meta-analyses [36]. It is noteworthy that plasma omentin concentration was decreased in patients with normal BMI and T2DM compared to the control group [32].

An in vitro study showed that omentin increases the transmission of signals related to insulin action by activating protein kinase B (Akt) and intensifying the adipose tissue response to insulin [5]. Therefore, it seems reasonable to suspect that a reduced concentration of this adipokine in plasma may be a component of the mechanism of insulin insensitivity in the human body [32].

A significant reduction in plasma omentin concentration was observed in patients with T2DM complicated by microcirculation damage conditions, such as diabetic retinopathy (DR), diabetic neuropathy, and diabetic nephropathy [37]. Attention was also drawn to plasma omentin concentration difference in patients suffering from DR and patients with T2DM only [38]. It may indicate a meaningful role of omentin as an antiangiogenic factor responsible for reducing synthesis of vascular endothelial growth factor (VEGF) and consequently opposing development of DR [39].

It is worth noting that a similar correlation as between omentin and T2DM cannot be found in patients with type 1 diabetes mellitus (T1DN) [36]. Some studies suggest a decrease in omentin plasma concentration, while others describe an increase in its concentration [36]. It is indeed a curious direction for further studies, which may explain the role of omentin in the pathogenesis of diabetes mellitus.

Cited studies show a relationship between the synthesis and expression of omentin and elements of metabolic syndrome—insulin resistance, T2DM, and obesity. Other, equally interesting observations were made regarding its components related to the cardiovascular system.

#### 2.2.3. Cardiovascular System, Lipid Metabolism, and Atherosclerosis

Omentin appears to have a very broad influence on the proper functioning of the circulatory system. It is proven that omentin has a protective effect on smooth muscle calcification in the arterial wall via the PI3K/Akt pathway [40] and a significant vasodilating effect on arteries. This effect is indirectly caused by increasing the synthesis of NO in endothelial cells, which has been studied in vitro in material collected from rats [26] and in the bovine aortic endothelial cells (BAECs) line [41]. Undoubtedly, it would be highly beneficial to carry out a similar study on human material.

Kazama et al. also made some interesting observations. The researchers noted that the administration of omentin significantly decreases blood pressure in rats with pharmacologically induced hypertension using NE, angiotensin II, and dimorpholamine [42]. In turn, another study showed a reduction in pulse pressure (PP), mean arterial pressure (MAP), and heart rate (HR) in normotensive rats [43]. This may be due to the omentin influence on increasing NO synthesis in blood vessels and reducing the expression of the interleukin-6 gene in pericardial adipose tissue [43]. The influence of omentin on blood vessels and blood pressure and its anti-inflammatory effect gives hope for the future use of reduced plasma concentration of this hormone as a biomarker of epithelial damage, among others, in the development of arterial hypertension [44].

Omentin also has a significant impact on the body’s lipid metabolism [45]. Furthermore, plasma omentin concentration may be related to diet. An increase in the concentration of this hormone was observed in people who follow a low-fat diet rich in monounsaturated fatty acids [46]. Plasma concentration of HDL was positively correlated with plasma omentin concentration. On the other hand, there was a negative correlation between this hormone concentration and plasma concentration of very low-density lipoprotein (VLDL) [14].

Importantly, low omentin concentrations promote the synthesis of triglycerides in hepatocytes, which can limit the transformation of VLDL into HDL, thus explaining the relationship between omentin concentrations and VLDL and HDL concentrations. This is an undesirable condition that can contribute to the development of atherosclerosis [47].

In another study, omentin was shown to promote macrophage differentiation into the anti-inflammatory M2 phenotype and suppress oxLDL-induced foam cell formation in macrophages [48]. Moreover, omentin appears to affect total cholesterol levels by increasing the number of receptors for low-density protein (LDL) on hepatocytes and reducing the amount of cholesterol reabsorbed by the intestine [48]. Lower omentin concentrations were also observed in patients with advanced atherosclerosis and T2DM compared to patients struggling only with T2DM [49]. All these factors may indicate a significant association of omentin with hyperlipidaemia and hypercholesterolemia and a potential vasoprotective effect of this adipokine.

#### 2.2.4. Atherosclerotic Plaque Rupture and Ischemic Stroke’s Consequences

The association of omentin concentration with atherosclerotic plaque formation corresponds to the data collected in the extensive prospective cohort study by Xu et al. Researchers linked plasma omentin levels to the prognosis of ischemic stroke (IS), which is a consequence of, among other things, atherosclerotic plaque rupture [50]. After analyzing 266 cases, it was found that patients with lower omentin levels were less likely to undergo IS without significant functional impairment [51]. The same researchers in a later paper described an association between lower omentin levels and higher mortality in patients with IS within one year of the event [52]. Also noteworthy is the negative correlation between omentin concentration and atherosclerotic plaque stability [53]. These data provide a strong indication of the potential use of omentin as a biomarker in patients with IS.

#### 2.2.5. Coronary Heart Disease, Intravascular Intervention, and Patient’s Prognosis

There were also attempts to link the action of omentin and the progression of coronary heart disease. Preliminary studies suggested a negative association between omentin levels and the exacerbation of coronary artery disease [54]. However, as the authors themselves mentioned, more clinical trials are needed to confirm these premises.

The determination of omentin concentration at the current level of knowledge does not allow its use in clinical practice. This does not change the fact that this adipokine will most likely be used as a biomarker of the risk of developing metabolic syndrome elements and its complications. This also applies to the assessment of patients’ prognosis during treatment. For example, in patients undergoing intravascular intervention, a correlation between a lower omentin concentration and a higher risk of complications from the procedure was shown [55]. In turn, in people struggling only with hypertension, the development of metabolic syndrome in the future can be predicted based on omentin [56]. However, despite the growing interest of scientists in the subject, many studies still need to be conducted to fully understand the relationship and role of omentin in the pathogenesis and course of metabolic syndrome.

### 2.3. Omentin in Wider Context

Primarily, omentin was mostly associated with metabolic syndrome and its comorbidities. In the course of the last two decades, this approach has been revised; thus, the direction of research has changed towards looking for correlations between omentin and other illnesses. Currently, a great deal of scientists focus on finding out omentin’s role in various inflammatory diseases.

It should be noted that no omentin receptors have yet been identified in the human body, despite our knowledge of various omentin’s signaling pathways. The first identified path was AMPK phosphorylation in adipocytes [5]. Nowadays, the most popular direction of molecular research is studying endothelial cells, due to which it is known that the AMPK phosphorylation is not the only pathway influenced by omentin. Other phosphorylated proteins are Akt and eNOS; moreover, an increase in PPAR δ expression is observed [57], which contributes to omentin’s positive effect on blood vessel condition. It was shown that omentin plays a role in the inhibition of the proinflammatory route TXNIP/NLRP3 in adipose tissue [58]. PI3K, ERK, and AMPK pathways are involved in the joint’s fibroblast function [59]. Omentin increases expression of mitochondrial biogenesis factors PGC-α and NRF-1 [60,61]; however, it also decreases IRF-1 expression through inhibiting the JAK2/STAT3 pathway in chondroblasts [62]. Omentin reduces differentiation of osteoblasts and consequently limits formation of osteoclasts in vitro due to lower RANKL levels [63]. It was reported that omentin interacts with transcription factors EN1, NFIC, ELK4, GATA2, JUND, FOXA1, and YY1, which are linked to the formation of a vast number of tumors [64]. Omentin in human hepatocellular carcinoma cells stabilizes proteins p53 and p21 and influences the JNK pathway, which overall has a toxic effect on the tumor [65]. In the liver’s macrophages, omentin decreases the activity of the mTOR pathway, which has a positive impact on this organ [66]. In kidneys, omentin down-regulates the expression of inflammatory cytokines, such as TNF-α or IFN-γ, and levels of oxidative stress. This role, as well as the general functional improvement of kidneys, is probably mediated by a decrease in miR-27a gene expression, which inhibits Nrf2 from expressing in patients with type 2 diabetic nephropathy [67]. Omentin also intensifies Nrf2 expression in ulcerative colitis (UC) and CD [68]. It was reported that in human ovarian granulosa cells, omentin increases steroid synthesis through enhancing the cell’s sensitivity to insulin-like growth factor (IGF-1) [69]. It was also reported that omentin influences the development of cardiomyocytes by interacting with BMP7 [70].

Omentin’s correlactions with various conditions, mentioned in the following subsection, were summarised in Figure 1.

#### 2.3.1. Osteoarthritis—Omentin’s Anti-Inflammatory Role

It is a functional and structural disorder of joints as a result of chronic inflammation that often leads to permanent disability. It is a common problem for which there is no fully effective treatment method. For some time, osteoarthritis (OA) was associated with obesity [71], and neoterically a correlation between serum omentin concentration and severity of this disease in obese patients has been found [72]. Application of exogenous omentin decreases secretion of matrix metalloproteinase (MMP) in cartilage tissue, which inhibits its degradation and slows down the progress of OA [62]. It is also proven that omentin increases the means of production of the anti-inflammatory cytokine IL-4 in fibroblasts and reduces the synthesis of pro-inflammatory cytokines IL-1β, IL-6, IL-8, and TNF-α [59], which have a crucial role in the development of OA [73]. Additionally, omentin’s influence on macrophages’ behavior was noted. This adipokine alters, as stated earlier, their phenotype to M2 type, which secretes a lot more anti-inflammatory molecules, thus having an osteoprotective effect [59].

#### 2.3.2. Tumors’ Development and Oncological Patient’s Prognosis in Relation to Omentin

It is widely acknowledged that obesity is a risk factor in many tumors and that it worsens patients’ survival prognosis [74]. An interesting deviation from this rule is lung cancer, in which obesity reduces the risk of the disease and even improves prognosis among affected patients [75]. Because of this paradox, various adipokines in patients with lung cancer were analyzed, and only the omentin expression turned out to be significantly altered—it was decreased regardless of the patient’s BMI. Moreover, its high concentration correlated with a better course of the disease and increased survival rate [54].

Tumors highly differ in their forms and the symptoms they cause. The concentration of plasma omentin in the past research showed immense fluctuations—ranging from 2 ng/mL up to 1100 ng/mL [76]. Because of such diverse studies’ results [77], in 2022 a meta-analysis was conducted to define plasma omentin concentrations in different tumors [78]. In this research paper, the poor quality of most studies and the lack of standardized reference ranges for healthy and obese people were highlighted. However, it was noted that there are differences in plasma omentin concentration between patients with different tumors, without a clear tendency towards any side of the spectrum. It was explicitly proven that there is a correlation between the appearance of gastrointestinal tumors and high plasma omentin concentration [78,79,80,81,82,83,84], while reports on breast cancer indicate its co-occurrence with low plasma omentin concentration [78,85,86,87,88]. The meta-analysis also questioned the current view on women having naturally higher omentin expression levels than men. It was shown that omentin is at slightly lower concentration in individuals with BMI under 25 than in people with BMI above 25, which contradicts earlier studies [78]. These inconsistencies show the need for standardization of omentin research in order to enhance their credibility, which was also pointed out by Paval et al. (2024). Correlation between tumors and adipokines is hard to study, due to simultaneous correlations between obesity and adipokines and obesity and tumors [77,89]. Additionally, while omentin’s concentrations vary in different cancers, its exact role in tumor progression and immune interactions needs further investigation, as it may answer questions about both tumors and the discussed adipokine.

Ovarian cancer (OC) presents a curious ability to decrease omentin expression both in surrounding tissues and in the whole organism, which was confirmed by injecting mice with the OC cell line. Studies have shown a correlation between low omentin serum concentrations and worse prognosis for patients with OC [90,91]. Additionally, it is proven that treating OC with exogenous omentin reduces malignant infiltration abilities by decreasing MMP synthesis. This effect is presumably achieved by binding lactoferrin [90,92] and may be a clue on where to search for potential omentin receptors.

In addition to previously mentioned tumors, a correlation between high serum omentin levels and advanced clinical stage of gastric, colon, pancreatic tumors [93], and prostate cancer [78,94,95] was observed. It is proven that in endometrial cancer, omentin is down-regulated alongside the disease’s progression [78,91,96,97].

#### 2.3.3. Asthma and Atopic Dermatitis

Asthma is a disease characterized by allergic inflammation of the lungs with slowly progressing fibrosis, hyperreactivity of airways, and oversecretion of mucus [98]. A condition often associated with asthma is atopic dermatitis (AD) [99], which also has a chronic inflammation origin [100]. During the course of both diseases, IL-25, IL-33, and TSLP cytokines are notably increased, which are all substantial in provoking a pathological immunologic response, proven in a mouse asthma model [101,102,103], and TSLP alone in a mouse AD model [104]. Precise mechanisms behind these diseases remain unknown. Interestingly, pulmonary goblet cells excrete omentin [105], and as it was shown, mice with genetically modified decreased omentin expression are protected from induced asthma development to a great extent [106]. However, omentin also presents a protective role against fibrosis [107]. The study conducted on people whose omentin expression in lungs was suppressed using nonsense mRNA showed that the synthesis of IL-25, IL-33, and TSLP was consequently decreased, directly linking omentin’s role in their production and, therefore, in the asthma development [106]. The proinflammatory role of omentin associated with TSLP and IL-33 was also shown in AD [106]. Mentioned mechanisms were confirmed in people without outside intervention, as well as increased omentin expression in these conditions [106,108]. Omentin and its correlation with other pulmonary diseases were described in detail by Zhou et al. (2017) [109].

#### 2.3.4. Sepsis and Septic Shock

It is an emergency condition caused by the whole organism’s hyperreaction, usually in the course of bacterial infection, and can be life-threatening [110]. Plasma omentin concentrations were analyzed in patients with sepsis and in septic shock. Unfortunately, there are no consistent or conclusive studies comparing plasma omentin concentration between patients in sepsis and healthy control trials [111,112]. Some studies suggest high serum omentin levels as a negative prognosis factor [111,113], while others state it as a positive prognosis factor [112,114]. This is yet another manifestation of the current omentin research’s standardization problem. Omentin’s concentration dynamics or simple correlation in sepsis could be an interesting and valuable observation, explaining a little more about the regulation of this adipokine.

#### 2.3.5. Metabolic Dysfunction-Associated Steatotic Liver Disease

It is the most common form of chronic inflammatory liver disease—approximately 30% of people around the world are affected by it. Origins of its development are associated with obesity and diabetes as risk factors [115]. Metabolic dysfunction-associated steatotic liver disease (MASLD), formerly known as non-alcoholic fatty liver disease (NAFLD), can lead to hepatocirrhosis, hepatocarcinogenesis, or even liver failure [116,117]. Research on omentin in patients with MASLD was controversial and ambiguous from the very beginning. In the first study, elevated serum omentin concentrations in patients with MASLD were observed [25], which did not match past views considering low omentin levels in conditions linked to obesity and T2DM. Recent meta-analyses did not manage to reach global consensus, but there is a slight research prevalence towards plasma omentin concentrations being lowered in MASLD [118]. Interestingly, various studies showed a diversity between populations. Namely, the Asian population with MASLD had reduced plasma omentin concentrations; in the Middle East, these values were reported to be increased, and for Europeans, there was no statistically significant difference between ill patients and healthy controls [118,119]. Although these dissimilarities may possibly be an effect of different medical procedures and norms in each region, at this point we cannot eliminate ethnic affiliation influence, especially since the prevalence and severity of MASLD vary between continents. Differences in general omentin’s plasma concentrations are a noticeable problem. Depending on a region, they are oscillating between 4 ng/mL and 400 ng/mL [118]. This data are presented in Table 1. to visualize the issue. This omentin-related phenomenon, similar to that observed in tumor research, strongly suggests that science needs methodology unification in order to validate performed studies.

Omentin, as a factor increasing adipocyte response to insulin, should have a protective role against MASLD [5]. Research showed that it, indeed, had a hepatoprotective role, however, it is due to preserving autophagy abilities in the liver’s macrophages by normalization of the AMPK/mTOR pathway [66].

#### 2.3.6. Crohn’s Disease and Ulcerative Colitis

CD is an inflammatory disease of the whole gastrointestinal system, from oral cavity to anus, but in most cases it expands throughout the intestines, starting with the ileocaecal fragment. Its aetiology remains unknown [132]. In an active CD, omentin is substantially lowered and proven to be a better marker of a disease stage than CRP. Additionally, omentin’s level of expression is further reduced in tissue affected by inflammation in contrast to unoccupied parts. Interestingly, in the CD’s remission state, plasma omentin concentration is comparable to a healthy person’s omentin concentration [68,133]. Similar observations were made in patients with UC [68,134]. Genetic and biotechnological studies did not find any significant changes in the amino acid sequence of omentin or some abnormalities in its function [9]. Experimentally, it was shown that injecting a mouse model of UC or CD with exogenous omentin ameliorates the severity of these conditions by influencing the Nrf2 pathway and secondarily NF-κB [68].

#### 2.3.7. Polycystic Ovary Syndrome

PCOS is a very common endocrine pathology in women of reproductive age, occurring in up to 15% of said population. It is a cause of many infertility cases and irregular menstruations [135]. Insulin resistance plays a key role in this syndrome and is associated with reduced levels of the omentin gene’s expression [136]. Even though in PCOS alone, in patients with both normal and high BMI, omentin reduced concentrations were observed [137], the statistical significance of this study was questioned by a meta-analysis [138]. It highlights the need for standardization of omentin studies. In healthy controls, plasma omentin concentration is similar to omentin’s concentration in follicular fluid, and what’s interesting is that in PCOS its levels are higher in follicular fluid in comparison to plasma concentration [69]. It was also reported that plasma omentin concentration is increased at the early stage of PCOS, probably functioning as an inhibitor of the acute phase [139].

### 2.4. Omentin—Other Correlated Diseases and Factors

#### 2.4.1. Omentin’s Concentration in Various Conditions

During the last few years, omentin was correlated with many other diseases, which were not as thoroughly studied, often due to small trials. These correlations are presented in the Table 2.

Some other studies about omentin suggested, that:Increased plasma omentin concentrations have moderating effect on psoriasis [152]Increased plasma omentin concentrations may have protective role in degeneration of nucleus pulposus in intervertebral discs [153]Plasma omentin concentrations demonstrates mixed effect on bone metabolism potentially reducing osteoporosis severity. However, its high concentrations are correlated with decrease in bone density in women with diabetes and older man [154]

#### 2.4.2. Omentin’s Effects in Animal Models

Throughout recent years, tests with animal participation were also greatly intensified. They proved an important point—omentin is not only a biomarker, but it also has its own impact on tissues and can be a potential therapeutic. These studies are showed in Table 3.

#### 2.4.3. Additional Factors Influencing Omentin’s Concentration

There was not much research analyzing omentin expression’s influence on various substances in a wider context. Past studies focused rather on the relation between omentin expression and the patient’s phenotype or lifestyle. So far it has been established that omentin expression increases after metformin treatment in women with PCOS [27] and after usage of birth control pills [177]. Its levels rise alongside increasing sex hormone-binding globulin (SHGB) concentration [178] and testosterone/SHGB ratio, while the rise of testosterone alone does not have any significant effects [96]. It was reported that statin therapy elevates the concentration of serum omentin [179,180]. Providing rats with ibuprofen increases their plasma omentin concentration [172]. The confirmed physical factor inhibiting omentin synthesis and secretion is hypoxia of adipose tissue [181].

## 3. Discussion

Until now, omentin has been called “a novel adipokine” despite being studied for the last two decades. Its role in metabolic syndrome has been established throughout this time. We know it plays a role in obesity and T2DM, but also in other diseases.

Obesity is a cause of many dangerous conditions, such as cardiovascular diseases, T2DM, chronic lung diseases, and increased risk of various cancers [29,31]. As long as obesity is a global problem, omentin will also remain an interesting research subject [30]. Its role in body metabolism and regulation of body mass has not yet been fully described; however, we know more about its function in obesity comorbidities every year. A special part in connecting these diseases and omentin is most likely played by anti- and pro-inflammatory factors [34], which always participate in the body’s complex balance and homeostasis. Spotting and understanding these correlations is very hard due to their reactionary nature and the fact that the mechanisms standing behind inflammation are yet to be fully explored. Identifying omentin’s role could help to better understand these mechanisms.

Current knowledge points out many simple correlations between plasma omentin concentrations and the course of diseases. Most of the time, high omentin levels are associated with a better health state in comparison to individuals with low levels. Examples of this tendency are obesity itself [14,32,33], T2DM [32,36], atherosclerosis [10,47], OA [72], PCOS [23,136,137], CD [7,68,133], UC [134], MASLD [118], OC [90,91], endometrial cancer [78,91,94,95], breast cancer [78,85,86,87,88], and lung cancer [54]. However, this general principle is not absolute and has some exceptions. Higher omentin concentrations can be associated with gastrointestinal tumors [78,79,80,81,82,83,84], prostate cancer [78,96,97], advancement of clinical stage of gastric, colon, and pancreatic tumors [93], as well as severity of some inflammatory diseases like asthma [106] or systemic sclerosis [150]. A lot of these conditions have unknown origins, and an analysis of omentin’s role in them could potentially help with filling these gaps. When it comes to tumors, the function of adipokines still needs to be further investigated, as their meaning could potentially extend their usage as a biomarker.

For a certain period of time, a lot of scientists were trying to answer the dilemma—does omentin play an active role in physiology, or is it only an accidental biomarker? Today it is quite a simple question, thanks to many in vivo and in vitro experiments. Omentin does have various functions presented in Figure 2, most of which are anti-inflammatory and protective. The first discovered ability of omentin is the ability to increase adipocyte cells’ insulin sensitivity [5]. This effect is probably the main factor in the general condition improvement of the obese mouse model with mediated omentins’ overexpression [58]. Other mechanisms include the reduction in pro-inflammatory molecule production, such as CRP, NF-κB [28], TNF-α, or IFN-γ [34]. Recently, more attention has been brought to omentin’s influence on macrophages, which seems to alter their phenotype toward more “tissue friendly” M2 type. In the liver, it plays a role in MASLD prevention [66]; in cartilage, it inhibits the progression of osteoarthritis [59]; and for blood vessels, it stops plaque formation [48]. In a lot of animal models of human diseases, omentin has been proven to wield impact on many different tissues in many different ways. Altogether, in the vast majority, its effects have a positive influence on a body, which makes it a really promising therapeutic agent in the treatment of numerous different conditions. More experiments with human participants are needed to see if it will meet the expectations or if it will be a fiasco like many others.

Lack of a defined distinctive omentin receptor stays intriguing, considering how many effects it has. Intelectin was proved to bind strongly with bacterial galactofuranosyl and some other carbohydrates to a smaller extent, but not to human saccharides [4]. The identity of omentin’s first site of action is crucial in understanding and for further exploration of this adipokine’s function. It could also lead to the identification of cross-reactions between various pro/anti-inflammatory cytokines. There are only a few propositions for omentins’ receptors. For example, in research on the role of omentin in OC, lactoferrin was pointed out for binding with this adipokine and therefore decreasing MMP synthesis [90,92]. Another way omentin acts is by connecting with integrin receptors αvβ3 and αvβ5 on macrophage cell membranes, by which it inhibits the expression of pro-inflammatory cytokines and the apoptosis process [165]. More studies focused on this subject are needed.

It is also worth noting that a huge part of the omentin’s research is conducted in vitro on animal tissue or animal models. While these studies are crucial to understanding molecular mechanisms and the relationship of omentin with various conditions, research on the human population is necessary to assert omentin’s practical properties. Appendix A presents studies conducted on humans that are discussed in this paper.

Unfortunately, research surrounding omentin has many issues. The lack of standardization is a huge problem and causes regress to some degree. Every original study has its own way of taking samples and has different measuring methodology. Possible differences affecting results are showed in Table 4. It leads to a situation where most meta-analyses have been pointing out a poor quality of experiments and had to narrow their search field to 5 works on average, despite the presence of a few dozen articles in medical libraries [78,118,138]. What is even more concerning is that these meta-analyses often challenge previous views, which were considered fundamental knowledge. For example, Lin et al. (2021) questioned if omentin is really down-regulated in PCOS [138]. Even basic matters, such as the correlation between low concentrations of omentin and obesity, were undermined by Paval et al. (2024) [78]. In this analysis, authors put special care into standardization and propose a mean omentin concentration for healthy controls—234 ± 21 ng/mL; then again, this data has not been widely validated. However, it was noticed that even in the general population, these levels can fluctuate and largely deviate [78]. Moreover, omentin’s basal levels vary depending on a region, which also points out probable differences in methodology between countries and studies (Table 1). This often results in improbable oscillations between 4 and 400 ng/mL [118]. Another issue is that a lot of smaller research fields are struggling with full polarization of results. As in sepsis, for example, where half of the studies implied omentin to be a positive prognostic factor [112,114], whilst the other half implied that it is a negative prognostic factor [111,113].

Without more standardized high quality studies research cannot go further, as a major part of current state of knowledge could turn out to be misleading or even worse—false, resulting in information chaos and unnecessary pause of the omentin research progress.

## 4. Conclusions

Up to this day, omentin remains an interesting direction for research. Its function and role are linked to many human conditions and diseases, such as obesity, diabetes mellitus, atherogenesis, lung problems, syndromes associated with the liver, ovaries, intestines, and osteoarticular system. However, there are still many aspects of omentin to explain and further investigate, especially in inflammatory diseases and tumors. Although omentin’s higher concentrations are usually correlated with better health states, it is not always the case. This adipokine is not only an interesting biomarker but also a potential therapeutic substance, which was proven in many studies using animal models. Unfortunately, there are many technical issues in omentin’s research field, primarily the lack of standardization between studies, which leads to many contradictory results denying each other and the inability to revise said contradictions.

## Figures and Tables

**Figure 1 biomedicines-13-00632-f001:**
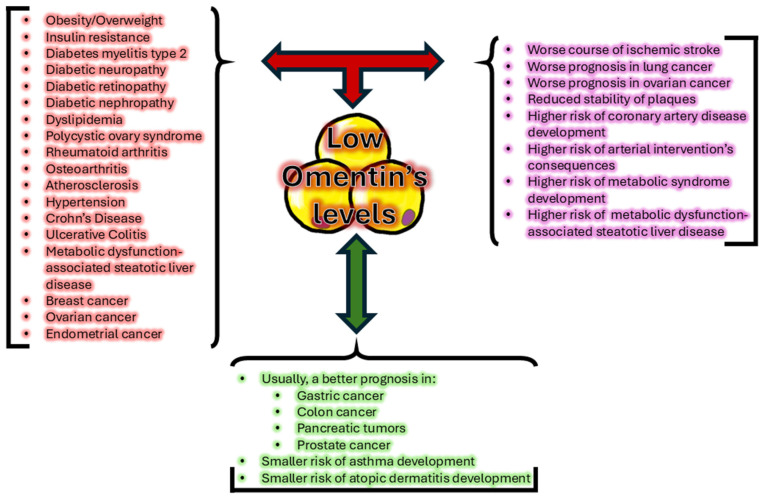
Brief summary of conditions, diseases, and prognosis correlated with low plasma omentin’s levels.

**Figure 2 biomedicines-13-00632-f002:**
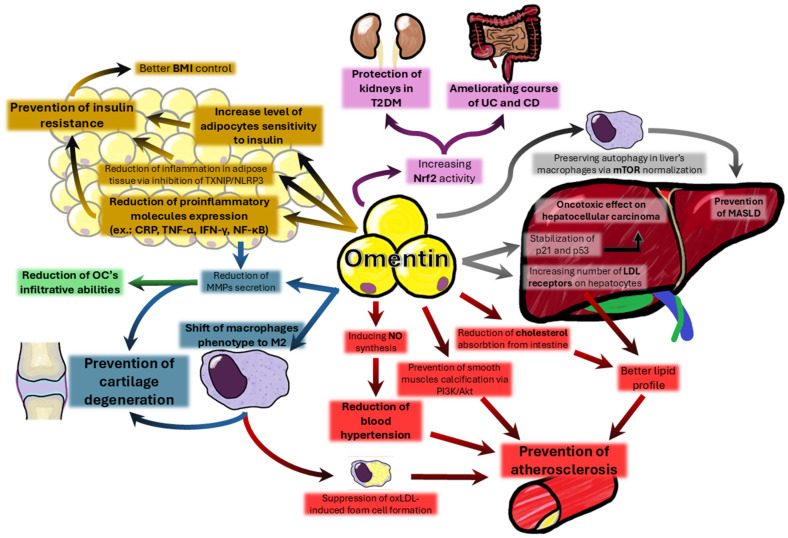
Summary of selected important functions of omentin. T2DM—type 2 diabetes myelitis; UC—ulcerative colitis; CD—Crohn’s disease; BMI—body’s mass index; CRP—C-reactive protein; TNF-α—tumor necrosis factor alpha; IFN-γ—interferon gamma; MMPs matrix metalloproteinases; OC—ovarian cancer; NO—nitric oxide; LDL—low-density lipoprotein; MASLD—metabolic dysfunction-associated steatotic liver disease.

**Table 1 biomedicines-13-00632-t001:** Differences in the omentin plasma concentrations in patients affected by MASLD between different regions, considering factors such as age and sex of participants. The presented data was based on the meta-analysis by Zhang et al. (2023) [118] and simplified to better show said differences.

Region	Study (Subregion) Age, Sex (% of Males)	Control Group Omentin Concentrations (ng/mL)	MASLD Group Omentin Concentrations (ng/mL)
China	Hang et al. [120] 45 ± 7; 63%	66.5 ± 8	44 ± 10
	Wang [121] 60 ± 10; 43% MASLD, 57% control	90 ± 18	
	Liu et al. [122] 65 ± 6; 82%	46 ± 6	36 ± 6
	Yu et al. [123] 65 ± 6; 100%		
	Zhang and Di [124] No data		
	Li et al. [125] 59 ± 10; 50%	23 ± 3	26 ± 4 18 ± 3.5 *
	Shao et al. [126] 58 ± 10; 46%	36 ± 3	
	Wang and Wang [127] 43 ± 7; 38%	4.5 ± 4.5	3 ± 5
Europe	Bekaert et al. [128] (Germany) 45 ± 11; 68% MASLD, 100% control	409 ± 124	393 ± 131
	Waluga et al. [129] (Poland) 31 ± 10 MASLD, 42 ± 15 control; 54%	114.5 ± 96	266.5 ± 82.5
Middle East	Yilmaz [130] (Turkey) 48 ± 8; 50%	376 ± 196	460 ± 181
	Montazerifar et al. [131] (Iran) 38 ± 8; 31% MASLD, 41% control	3 ± 4.5	3.4 ± 5.5 **

* patients with MASLD and T2DM, conducted by Li et al. ** patients with MASLD and abdominal obesity.

**Table 2 biomedicines-13-00632-t002:** Diseases correlated with omentin’s concentrations in recent studies. Omentin’s concentrations were simplified.

Disease or Condition	Age, Sex (% of Males), Ethnicity	Omentin’s Plasma Average Concentrations in Study Group	Additional Notes
Primary nephrotic syndrome [140]	4 ± 3, 70% Chinese	↘ Reduced ↘ 159 ng/mL	Reduced levels in active stage as well as in remission [140]
Obstructive sleep apnoea [141]	44 ± 13, - Chinese	↘ Reduced ↘ 18 ± 8 ng/mL 250 ± 30 ng/mL	General concentrations were varying between studies, therefore given results are grouped
	46 ± 13, 71% Turkish	↘ Reduced ↘ 26 ± 7 ng/mL	
Central precocious puberty in girls [142]	7 ± 1, 0% Turkish	↘ Reduced ↘ 31 ± 10 ng/L	-
Irritable bowel syndrome [143]	11 ± 6, 66% Polish	↘ Reduced ↘ 300 ± 20 ng/mL	-
Periodontitis [144]	47 ± 13, - Indian	↘ Reduced ↘ 185 ng/mL 490 ng/mL *	Anti-inflammatory role in vivo model of said disease [145]
Hepatitis C virus infection [146]	63 ± 20, 57% German	Unchanged 20 ± 10 ng/mL	Concentration is increased when hepatocirrhosis occurs [146]
Kawasaki disease [147]	3 ± 1, 53% Chinese	↗ Increased ↗ 36 ± 5 ng/mL	-
After tibial cortex transverse transport in treatment of diabetic foot [148]	-, 61% Chinese	↗ Increased ↗ 40 ± 10 ng/mL	-
Dengue haemorrhagic fever [149]	27 ± 10, 59% Singhlese	↗ Increased ↗ 974 ± 200 ng/mL	-
Systemic sclerosis [150]	54 ± 13, 54% Polish	↗ Increased ↗ 556 ± 150 ng/mL	-
Inflammation of genitourinary system in men [151]	32 ± 7, 100% Italian	↗ Increased ↗ 10 ng/mL **	-
Impotency in men [151]		↗ Increased ↗ 5,2 ng/mL **	-

* Reduced concentrations were observed in saliva. ** Increased concentrations were observed in sperm.; “-”—No data.

**Table 3 biomedicines-13-00632-t003:** Discovered effects of exogenous omentin or its physiological role in various animal models.

Effect or Role of Omentin	Used Animal Model
Reduces pulmonary hypertension	Mouse [155]
Reduces blood pressure in preeclampsia	Mouse [156]
Presents positive effects on thrombolytic therapy after stroke	Mouse [157,158]
Prevents mild prostate hypertrophy	Mouse [159]
Reduces osteoporosis	Mouse [160,161]
Participates in the proper cardiomyocytogenesis	Mouse [70]
Improves reconstruction of cardiomyocytes after heart ischemia	Mouse [162,163,164]
Prevents development of atherosclerosis or potential rupture of already existing plaque	Mouse [165,166] Rat [167]
Prevents development of aneurysms in hypertension induced by apo-E knock-out	Mouse [168]
Have moderating effect on symptoms and damage in UC	Mouse [169]
Down-regulating slows down bone healing	Mouse [170]
Down-regulating prevents development of inducted asthma	Mouse [106]
High levels seems to speed up development of RA and to be correlated with severity of said disease	Mouse [171]
Reduces cardiovascular complications without influencing blood pressure itself in spontaneous hypertension	Rat [172]
Reduces blood pressure in inducted hypertension	Rat [42]
Reduces blood pressure in normotension and slows down heart rate	Rat [43]
Improves general function of vessels in T2DM, takes out many oxidative particles and also reduces blood pressure through ameliorating insulin resistance in blood vessels	Rat [173,174]
Reduces scarification in damaged vessels in rat [163]	Rat [175]
Protects joints from degeneration in inducted osteoarthritis [59]	Rat [59]
Could play a role in menstruation cycle, as its expression in hypothalamus varies depending on menstrual time; its expression is stimulated by gonadotropins and gonadotropin-releasing hormone	Pig [176]

**Table 4 biomedicines-13-00632-t004:** Examples of important standardization issues causing difficulties in comparing results across studies.

Issue	Description
Blood sampling protocol	There is no determined protocol for collecting blood from patients for omentin’s plasma concentration test. This includes time of collecting blood samples or fasting/non-fasting state. Medicaments used by patients should also be taken into consideration as well as their overall condition.
Elisa kits	Elisa kits designed to detect omentin could vary in-between different manufacturers. Each test often has unique antibodies that can differ in affinity or even calibration methods. This may result in inconsistencies among studies.
Storage conditions	Way of storing samples can affect their quality. Omentin is a protein, so such factors as temperature, presence of peptidase inhibitors or time of storage may lead to changes in its concentration or its detection rate.
Preparation of samples	Many analytical procedures have their own reference range due to various methods, despite being in use for many years. However, omentin does not have neither widely validated and accepted concentrations norms nor standardized methods of testing. This probably results in many differences and big fluctuations between various researches.

## Data Availability

Documents containing all extracted data are available in the manuscript.

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
