# Peer review of "Omentin—General Overview of Its Role in Obesity, Metabolic Syndrome and Other Diseases; Problem of Current Research State"

_biomedicines, 2025, doi:10.3390/biomedicines13030632_

Round 1
Reviewer 1 Report
Comments and Suggestions for Authors
Numerous studies are currently focused on exploring the involvement of omentin under normal and pathologic conditions. As a result, the topic of the manuscript is interesting. However, to considerably improve the quality of the manuscript, I have some recommendations for the authors:
- Please revise the abstract content. Methods are insufficiently described (What means interesting and important articles? What are the criteria for inclusion? "Interesting" cannot be an inclusion criterion. Are there exclusion criteria? Is it a systematic review manuscript? A systematic review should strictly follow the PRISMA checklist and include a completed PRISMA flowchart in the main text or supplementary materials. A review manuscript structure may include an introduction, relevant sections, discussion, conclusions and future perspectives.
- The abstract should be approximately 250 words as recommended by Biomedicines.
- The methods section is unnecessary in a review-type manuscript.
- All subsections should be revised according to the fluency and content of the data presented (except 3.2.3. Cardiovascular system, lipid metabolism and atherosclerosis, 3.3 Omentin in wider context, Asthma and atopic dermatitis, Crohn’s disease and ulcerative colitis, and Metabolic dysfunction-associated steatotic liver disease subsections). I also suggest adding some statistics results in different situations (such as correlations or no correlations)
- Sections 3.3.1 and 33.2 should be organized in a table.
- Please review the English. Some sentences are difficult to read. (e.g. Various studies identified the in vitro effects of omentin and differences in its levels was observed depending on the patient's condition – lines 27-28; Omentin does not, however, increase base cell glucose uptake in contrast to many many other adipokines – lines 31-32; ’Nevertheless, no significant correlation has been demonstrated so far’ – with type 2 diabetes mellitus? Lines 37).
- Please add a reference in lines 14, 21 (page 3) and lines 6, 43 (page 4)
- Kindly add the study's future perspectives and the research limits.
- Please organize a Conclusion section of the manuscript.
- Please revise the reference list according to Biomedicines Journal recommendations.
Comments on the Quality of English LanguagePlease review the English. Some sentences are difficult to read. (e.g. Various studies identified the in vitro effects of omentin and differences in its levels was observed depending on the patient's condition – lines 27-28; Omentin does not, however, increase base cell glucose uptake in contrast to many many other adipokines – lines 31-32; ’Nevertheless, no significant correlation has been demonstrated so far’ – with type 2 diabetes mellitus? Lines 37).
Author Response
Reviewer 1
Comment 1: Please revise the abstract content. Methods are insufficiently described (What means interesting and important articles? What are the criteria for inclusion? "Interesting" cannot be an inclusion criterion. Are there exclusion criteria? Is it a systematic review manuscript? A systematic review should strictly follow the PRISMA checklist and include a completed PRISMA flowchart in the main text or supplementary materials. A review manuscript structure may include an introduction, relevant sections, discussion, conclusions and future perspectives.
Response 1: As requested, we revised the abstract content. We admit that “interesting” was not an adequately precise term and therefore we specified it. We believe that inclusion criteria are now clarified. We did not apply exclusion criteria, for as long, as the quality and relevancy of study was appropriate. This work is a literature review, not a systematic review.
Change made in response to this comment, can be found marked as red on page 1 lines 12-16: “Methods: MEDLINE and SCOPUS databases were searched using keywords “omentin” or “intelectin-1”. Then the most recent articles providing new perspectives on the matter and the most important studies which revealed crucial insight were selected to summarize the current knowledge on the role of omentin in a literature review.”
Comment 2: The abstract should be approximately 250 words as recommended by Biomedicines.
Response 2: After modifying the abstract according to instructions it has now 256 words. We believe that it is in line with the guideline of “approximately 250 words”. If you require any further modifications in this regard please kindly let us know.
Comment 3: The methods section is unnecessary in a review-type manuscript.
Response 3: Thank you for pointing this out. We were unsure, if this section would be necessary, which is why we included it in the article in the first place. Therefore, we deleted it from the work as recommended.
Comment 4: All subsections should be revised according to the fluency and content of the data presented (except 3.2.3. Cardiovascular system, lipid metabolism and atherosclerosis, 3.3 Omentin in wider context, Asthma and atopic dermatitis, Crohn’s disease and ulcerative colitis, and Metabolic dysfunction-associated steatotic liver disease subsections). I also suggest adding some statistics results in different situations (such as correlations or no correlations)
Response 4: We agree some of subsections titles could better reflect on its content. Thus, we altered the titles of subsubsections 2,1, 2.2.1, 2.2.2, 2.2.4, 2.2.5, 2.3.1, 2.3.2, 2.3.4, 2.4.3. Regarding statistics, we included them where evidently relevant to secure that too much numerical data does not disrupt the reading flow of this review.
Comment 5: Sections 3.3.1 and 3.3.2 should be organized in a table.
Response 5: We fully agree that a table in this part will be more attractive to read and easier to find the information. Therefore, we organized sections 2.4.1 and 2.4.2 (previously 3.3.1 and 3.3.2) in tables 1 and 3, which can be found on pages 9, 10 and 11.
Comment 6: Please review the English. Some sentences are difficult to read. (e.g. Various studies identified the in vitro effects of omentin and differences in its levels was observed depending on the patient's condition – lines 27-28; Omentin does not, however, increase base cell glucose uptake in contrast to many many other adipokines – lines 31-32; ’Nevertheless, no significant correlation has been demonstrated so far’ – with type 2 diabetes mellitus? Lines 37).
Response 6: Thank you for pointing this out. We reviewed the English again to meet Journal’s expectations and express the mentioned sentences more clearly. Lines 24-25 (previously 27-28) were changed to “Various studies identified in vitro effects of omentin and observed differences in its levels depending on the patient's condition.”; lines 28-29 (previously 31-32) “However, omentin does not increase basal cell glucose uptake, in contrast to many other adipokines.”; line 33 (previously 37) “Nevertheless, no significant correlation between these genes and T2DM has been demonstrated so far”. All of changes are localised on page 2.”
Comment 7: Please add a reference in lines 14, 21 (page 3) and lines 6, 43 (page 4)
Response 7: As suggested, we added references accordingly to the request. They can be found in lines 10, 17 on page 3 and lines 4, 41 on page 4.
Comment 8: Kindly add the study's future perspectives and the research limits.
Response 8: Regarding omentin study future perspectives we highlight omentin’s possible future as a biomarker and a potential drug (please refer to lines 12-13, page 5; lines 19-29, page 5; lines 28-29, page 12 etc.). We also state the need for future studies and difficulties associated with them. Our team is willing to follow the future reports about omentin and conduct original research on this topic. So far we did not encounter any major limitation of this study. If you have any further suggestions regarding this part, please let us know.
Comment 9: Please organize a Conclusion section of the manuscript.
Response 9: Thank you for suggesting this - we agree that the conclusions section will be beneficial for the article. Therefore, we organized it as requested in section 4 on page 19.
Comment 10: Please revise the reference list according to Biomedicines Journal recommendations.
Response 10: The reference list has been revised to follow Biomedicines Journal recommendations. All references were prepared using PubMed citation in NLM format.
Reviewer 2 Report
Comments and Suggestions for Authors
While the review provides a comprehensive summary of Omentin-1’s functions, it has several concerns that need to be addressed to enhance its clinical applicability and value:
- There is inconsistency in research methods, reference ranges, and sample collection, leading to difficulties in comparing results across studies. The authors should described Differences in Kits:ELISA kits from different manufacturers may use different antibody clones or calibration standards, leading to significant variations in results when testing the same sample in different laboratories. For example, the Omentin-1 concentration in healthy populations varies from 2 ng/mL to 1100 ng/mL across different studies. There is no standardized protocol for blood sample collection (fasting/non-fasting), storage conditions (freezing temperatures, repeated freeze-thaw cycles), and centrifugation speed, all of which can affect the test results. Currently, there is no widely accepted reference range for Omentin-1 concentration in healthy populations. Some studies suggest a mean of 234 ± 21 ng/mL in healthy individuals, but this data has not been widely validated. I suggest the authors add a table to describe these differences.
- Most studies are based on animal models or small clinical trials, with few large-scale studies in human diseases. Thus, I suggest add a table to summarize its results. Differences in Omentin-1 levels across ethnic groups, Gender and Age, especially in MASLD and other diseases, which are not discussed in depth.
- There is insufficient discussion on how Omentin-1 interacts with other adipokines like adiponectin and leptin.
- While Omentin-1’s concentrations vary in different cancers, its exact role in tumor progression and immune interactions needs further investigation.
- It should provide 3 pictures to briefly describe its role/mechanisms or problem in the current research, respectively.
There are several mistales in english grammar.
For example:
-
Original: "Omentin has been gaining popularity as a research topic over the last 20 years. The interest in it has increased substantially since its connection with numerous diseases... has been revealed."
Issue: The tense in the second half of the sentence is inconsistent ("has increased" and "has been revealed" creates unclear time logic). -
Original: "Omentin is adipokine, that is a peptide hormone..."
Issue: The article "an" is missing before "adipokine." -
Original: "It is a cause of many infertility and irregular menstruation cases."
Issue: "Many infertility" is incorrect as "infertility" is an uncountable noun. -
Original: "Omentin prevents angiogenesis in vitro [27] and has an anti-inflammatory effect, due to reducing c-reactive protein’s (CRP) and NFκB expression [28]."
Issue: "Due to" should be followed by a noun phrase, not a verb. -
Original: "In 2022 approximately 43% of all adults worldwide were struggling with overweight and 26% with obesity. These statistics are even less favourable in developed countries, for instance in the United States as much as 67% of its population is overweight."
Issue: A comma is missing after "for instance," and the sentence structure is loose.
6.Original: "It was proven to take part in development of asthma and atopic dermatitis."
Issue: "Take part in" is informal; a more formal expression is preferred in academic writing.
7.Original: "Omentin was proven to prevent osteoarthritis, hepatocirrhosis and atherosclerosis in mouse models, which places omentin among potential therapeutic products, and not only as a biomarker."
Issue: The sentence is overly long, and the logical focus is unclear.
Author Response
Reviewer 2
Comment 1: There is inconsistency in research methods, reference ranges, and sample collection, leading to difficulties in comparing results across studies. The authors should described Differences in Kits:ELISA kits from different manufacturers may use different antibody clones or calibration standards, leading to significant variations in results when testing the same sample in different laboratories. For example, the Omentin-1 concentration in healthy populations varies from 2 ng/mL to 1100 ng/mL across different studies. There is no standardized protocol for blood sample collection (fasting/non-fasting), storage conditions (freezing temperatures, repeated freeze-thaw cycles), and centrifugation speed, all of which can affect the test results. Currently, there is no widely accepted reference range for Omentin-1 concentration in healthy populations. Some studies suggest a mean of 234 ± 21 ng/mL in healthy individuals, but this data has not been widely validated. I suggest the authors add a table to describe these differences.
Response 1: Thank you very much for this comprehensive comment – fully agreed. We created Table 3 on page 14 which summarizes the omentin research standardization problem. We hope it addresses your point . If you have any additional suggestions, please kindly let us know.
Comment 2: Most studies are based on animal models or small clinical trials, with few large-scale studies in human diseases. Thus, I suggest add a table to summarize its results. Differences in Omentin-1 levels across ethnic groups, Gender and Age, especially in MASLD and other diseases, which are not discussed in depth.
Response 2: We agree that a table summarising results of meta-analyses and studies conducted on human population would greatly benefit our work. Therefore, we created extensive Table number 4 on pages 14,15,16, 17,18 and 19, and a short paragraph highlighting the need for research on human material on page 13, lines 15-19.
However, we refrained from going into details considering gender, age and ethnicity, as we tried to avoid overloading the work with excessive numerical information. If you still believe this data will add value to the article – please let us know. In fact, to properly discuss the mentioned differences, a separate meta-analysis would be required, which may be an interesting idea for a separate research paper in the future.
Comment 3: There is insufficient discussion on how Omentin-1 interacts with other adipokines like adiponectin and leptin.
Response 3: We would like to take a deeper look at interactions between omentin and other adipokines. However, it seems to be beyond our reach at the moment. These mechanisms remain largely unexplored and unexplained, and describing them in a satisfactory manner requires more original research as a base for review.
Comment 4: While Omentin-1’s concentrations vary in different cancers, its exact role in tumor progression and immune interactions needs further investigation.
Response 4: We agree that this research direction should have been highlighted in greater extent in our work. Therefore, we added a few sentences in “2.3.2 Tumors’ development and oncological patient’s prognosis in relation to omentin” subsubsection (previously named “3.3.2 Tumors”) and “3. Discussion” to point this out. Additions can be found on page 7, lines 11-14 (“Additionally, while omentin’s concentrations vary in different cancers, its exact role in tumor progression and immune interactions needs further investigation, as it may answer questions about both tumors and discussed adipokine.”) and on page 12, lines 12-14 (“When it comes to tumors, function of adipokines still need to be further investigated, as their meaning could potentially extent their usage as a biomarker”).
Comment 5: It should provide 3 pictures to briefly describe its role/mechanisms or problem in the current research, respectively.
Response 5: As recommended, we prepared 2 schemes to briefly summarize omentin’s correlations and its function. Figures can be found on page 9 and on page 13. Regarding problems in the current research we did not prepare an additional scheme, since we believe that Table 3 covers the topic in a calera and comprehensive way.
Comments on the Quality of English Language:
There are several mistales in english grammar.
For example:
- Original: "Omentin has been gaining popularity as a research topic over the last 20 years. The interest in it has increased substantially since its connection with numerous diseases... has been revealed."
Issue: The tense in the second half of the sentence is inconsistent ("has increased" and "has been revealed" creates unclear time logic). - Original: "Omentin is adipokine, that is a peptide hormone..."
Issue: The article "an" is missing before "adipokine." - Original: "It is a cause of many infertility and irregular menstruation cases."
Issue: "Many infertility" is incorrect as "infertility" is an uncountable noun. - Original: "Omentin prevents angiogenesis in vitro [27] and has an anti-inflammatory effect, due to reducing c-reactive protein’s (CRP) and NFκB expression [28]."
Issue: "Due to" should be followed by a noun phrase, not a verb. - Original: "In 2022 approximately 43% of all adults worldwide were struggling with overweight and 26% with obesity. These statistics are even less favourable in developed countries, for instance in the United States as much as 67% of its population is overweight."
Issue: A comma is missing after "for instance," and the sentence structure is loose. - Original: "It was proven to take part in development of asthma and atopic dermatitis."
Issue: "Take part in" is informal; a more formal expression is preferred in academic writing. - Original: "Omentin was proven to prevent osteoarthritis, hepatocirrhosis and atherosclerosis in mouse models, which places omentin among potential therapeutic products, and not only as a biomarker."
Issue: The sentence is overly long, and the logical focus is unclear.
Response: Thank you for pointing out these mistakes, and naturally we corrected each one of them.
- Lines 17-19 page 2 - “Omentin has been gaining popularity as a research topic over the last 20 years. The interest in it has increased substantially since its connection with numerous diseases […] was”
- Line 4 page 2 - “Omentin is an adipokine, that is a peptide hormone […]”
- Line 3 page 9 - “It is a cause of many infertility cases and irregular menstruations.”
- Lines 42-44 page 2 - “Omentin prevents angiogenesis in vitro [27] and has an anti-inflammatory effect, due to reduced c-reactive protein’s (CRP) and NF-κB expression induced by omentin.
- Lines 8- 9 page 3 - “These statistics are even less favourable in developed countries, for instance, in the United States as much as 67% of its population is overweight.”
- Lines 22-24 page 1 - “It was proven to participate in development of asthma and atopic dermatitis […]”
- Lines 26-28 page 1 - “Omentin was proven to prevent osteoarthritis, hepatocirrhosis and atherosclerosis in mouse models. All of the above places omentin among potential therapeutic products, and not only as a biomarker.”
Round 2
Reviewer 1 Report
Comments and Suggestions for Authors
The authors have provided a more organized revision of the manuscript. However, I have a few suggestions for the authors:
- Kindly revise the reference list according to the Biomedicines journal recommendations (please follow the link: https://www.mdpi.com/journal/biomedicines/instructions).
- Kindly increase the typographical features of the Figures (e.g. font size) and add a list of all abbreviations used in their legends.
Author Response
Dear Reviewer
Comment 1: Kindly revise the reference list according to the Biomedicines journal recommendations (please follow the link: https://www.mdpi.com/journal/biomedicines/instructions).
Response 1: The reference list has been revised to follow Biomedicines Journal recommendations. In particular we changed the way in which many consecutive references were made (e.g. [4][6] was changed to [4,6] on page 2, line 8, [7][8][9] was changed to [7-9] on page 2, line 9, etc.). Additionally, we included only the year of publication, instead of full date in every reference in the References section, which is in line with the journal guidelines. We deleted additional identifiers from the References section such as PMID and PMCID leaving only Digital Object Identifiers requested by the journal.
If you have any further specific suggestions regarding this part, please kindly advise.
Comment 2: Kindly increase the typographical features of the Figures (e.g. font size) and add a list of all abbreviations used in their legends.
Response 2: As recommended, we increased the size of features in Figure 2 and added adequate abbreviations in its legend.
Reviewer 2 Report
Comments and Suggestions for Authors
1.Very good improvement in this version, However, as a reviewer, I insist Differences in Omentin-1 levels across ethnic groups, Gender and Age, especially in MASLD and other diseases, which are not discussed in depth. This data will add great value to the article , a system review instead of a separate meta-analysis would be appropriate.
Author Response
Reviewer 2
Comment 1: Very good improvement in this version, However, as a reviewer, I insist Differences in Omentin-1 levels across ethnic groups, Gender and Age, especially in MASLD and other diseases, which are not discussed in depth. This data will add great value to the article , a system review instead of a separate meta-analysis would be appropriate.
Response 1: As suggested, we created a new Table 1 on pages 8, 9, focusing on these differences in various studies of MASLD patients. A new part referring the data in Table 1 was added on page 8 lines 30-34 and page 15 lines 19-21. Moreover, we updated Table 2 (previously Table 1) to include age, sex, ethnic group of participants and omentin’s exact concentration, whenever this data was mentioned. Added content is marked in red. We hope these changes are going to meet your expectations.